# Saikosaponin A alleviates depressive-like behavior induced by reserpine in mice by regulating gut microflora and inflammatory responses

**Menglin Wang[1][☉], Haojun Li[1][☉], Wenjing Zhang[1], Li Zhang[1], Shun Wang[2], Miao Jia[1], Lu Jia[1], Yu Zhang[1], Haifei Gao[1,3], Xianwu Zhang[1]\*, Zhaohui Yin[1]\***

1 Inner Mongolia Mental Health Center (The Third Hospital of Inner Mongolia Autonomous Region, Brain Hospital of Inner Mongolia Autonomous Region), Hohhot, P. R. China, 2 College of Veterinary Medicine, Northeast Agricultural University, Harbin, P. R. China, 3 Inner Mongolia Medical University, Hohhot, P. R. China

☉ These authors contributed equally to this work.
\* yzh7078@163.com (ZY); xwxwabc@sina.com (XZ)

## Abstract

Saikosaponin A (SSA), a key ingredient of Chaihu-Shugan-San, has been shown to possess anti-inflammatory, antioxidant and antidepressant properties. Therefore, the present study aimed to investigate the potential mechanism of action and the effect of SSA on reserpine-induced depressive-like symptoms in mice. Establishing mouse model of depression using intraperitoneal injection of reserpine (RSP). Forced swimming test, tail suspension test and sucrose preference test were used to assess depression-like behavior in mice. The results showed that mice exposed to RSP not only showed weight loss and depressive behavior, but also elevated levels of IL-1β and TNF-α, as well as upregulated levels of reactive oxygen species (ROS) and lipid peroxides in the hippocampus. Detection of changes in the intestinal flora of mice using 16S rRNA, it was observed that the intestinal flora changed following SSA treatment. Not only was there an increase in the overall abundance of the intestinal microbiota, but there was also a significant downregulation of the *Firmicutes* and an up-regulation of the *Verrucomicrobia* at the phylum level. Furthermore, SSA treatment markedly improved depressive-like behavior induced by RSP, alleviated damage to the hippocampus, elevated levels of monoamine neurotransmitters, suppressed inflammatory factors in the hippocampus, reduced hippocampal oxidative stress, and restored gut microbiota disruption in RSP-induced mice. The findings propose that SSA has the potential to alleviate depressive symptoms in mice by enhancing monoamine neurotransmitter levels, suppressing hippocampal inflammation, and modifying gut microbial composition.

## 1. Introduction

Depression is a common disorder that involves a variety of related mood disorders, including sadness, loss, frustration and even anger, and these negative emotions last from one to several

**Data availability statement:** All relevant data are within the manuscript and its Supporting information files.

**Funding:** This project was supported by the funding programs, "Natural Science Foundation of Inner Mongolia, 2024QN08067, and High level Clinical Specialist Construction Project of Public Hospitals in the Capital Region of Inner Mongolia Autonomous Region 2023SGGZ0005. The funders had no role in study design, data collection and analysis, decision to publish, or preparation of the manuscript.

**Competing interests:** The authors have declared that no competing interests exist.

**Abbreviations:** SSA, Saikosaponin A; RSP, reserpine; ROS, reactive oxygen species; IL-1β, interleukin-1beta; TNF-α, tumor necrosis factor-α; 5-HT, 5-hydroxytryptamine; DA, dopamine; NE, norepinephrine; Bax, BCL2-associated X; CREB, cAMP-response element binding protein; BDNF, brain-derived neurotrophic factor; CG, control group; TST, tail suspension test; SPT, sucrose preference test; ELISA, Enzyme-Linked Immunosorbent Assay; HE, hematoxylin-eosin; FJB, Fluoro-Jade B; LPO, lipid peroxidation; TNFR1, tumor necrosis factor receptor-1; NF-κB, nuclear factor kappa-light-chain-enhancer of activated B cells; SAP, severe acute pancreatitis; SNARE, soluble N-ethylmaleimide-sensitive factor attachment protein receptor; Syt1, synaptotagmin-1

weeks [1]. Major depression is the third leading global cause of disease burden, according to data published by the World Health Organization, and predicts that it will probably reach the top by 2030 [2]. Affecting over 300 million individuals globally, depression leads to varying degrees of impairment in psychosocial functioning and quality of life, and worse still, long-term severe depression may be associated with suicide attempts [3]. Depression is a public health problem to be addressed, not just a disease.

The precise etiology of depression remains unclear and is often perceived as overly complex. For instance, the monoamine hypothesis has been widely used to explain the classical pathogenesis of depression, and that the repletion of monoamine neurotransmitters plays an essential role in mood regulation [4]. Based on the monoamine hypothesis, a variety of monoamine oxidase inhibitors and selective serotonin reuptake inhibitors have been extensively studied; these drugs can effectively increase levels of monoamines such as 5-hydroxytryptamine (5-HT), dopamine (DA) and norepinephrine (NE), which are significant in improving patients' quality of life and rebuilding their confidence in life [5,6]. However, there are some pharmacological effects of these drugs that need to further elucidation. Previous studies have shown that oral paroxetine significantly increases 5-HT levels in patients within a week, but in reality patients need to take paroxetine for several weeks or more to achieve a therapeutic effect, which seems to contradict the monoamine hypothesis [7]. In recent years, there has been a growing recognition that the gut microbiota may be a potentially relevant factor in the onset and progression of depression [8]. Some evidence shows that dysfunction of the microbiota-gut-brain axis may play an influential role in the pathogenesis of depression, which involves the enteric nervous system, the immune system, the blood-brain barrier, the hypothalamic-pituitary-adrenal axis, and the central nervous system [9]. This hypothesis has been substantiated by animal experiments, demonstrating that the transplantation of feces obtained from depressed humans or depressive-like mice to healthy mice resulted in a deficit of pleasure and depressive-like symptoms [10]. For individuals suffering from depression, the consumption of probiotics has the potential to restore balance to gut flora fluctuations and modulate the metabolic activity of gut microbes. The production of beneficial metabolites may contribute positively to the reestablishment of homeostatic equilibrium in these patients [11–13]. Additionally, natural compounds hold considerable promise as antidepressants, with complex mechanisms of action that may involve brain function, immunity, the hypothalamic-pituitary-adrenal axis and gut flora [14–16]. Despite significant advances in comprehending depression, there is no single theory or mechanism that entirely elucidates the various facets of this disorder. Such limitations have restricted research into antidepressant medication [17]. Traditional Chinese medicine, which has shown efficacy in the treatment of depression and associated clinical conditions, and explaining the mechanism of action of natural compounds in Chinese medicine may be an acceptable option for understanding the pathogenesis of depression.

Chaihu-Shugan-San is a traditional herbal remedy that previous studies have demonstrated to exhibit antidepressant-like effects in a variety of depression-like animal models [18,19]. Saikosaponin A (SSA), a triterpenoid saponin, is the main ingredient in this formulation and has a wide range of pharmacological activities including anti-inflammatory, antioxidant and anti-malignant effects [20,21]. Previous research has indicated that SSA intervention can suppress the manifestation of apoptotic markers like Bax and Caspase-3, enhance the activity of hippocampal neurons. This ultimately has a positive impact on ameliorating post-stroke depression-like behavior in rats [22]. In addition, Guo performed a proteomic-based evaluation of the antidepressant properties of SSA, which indicated proline transmembrane protein 2 has a crucial role in a rat depression model caused by chronic unpredictable mild stress [23]. The findings have established a significant correlation between SSA and depressive symptoms,

suggesting a potential role of SSA in the pathophysiology of depression. Nonetheless, the underlying mechanisms of SSA's impact on mood disorders require more in-depth investigation to fully elucidate its therapeutic potential and possible side effects.

Reserpine (RSP), an alkaloid extracted from the roots of the serpentine loosestrife plant, possesses antihypertensive properties. It is widely used in the design of animal models of depression and in the screening of potential antidepressants [24,25] Accordingly, our study delved into the molecular underpinnings of SSA's antidepressant efficacy within the RSP-induced depression-like paradigm in mice, with a particular focus on the modulation of intestinal microbiota by SSA. Our research offers novel insights regarding the antidepressant effect of SSA and provide a basis for further treatment of depression.

## 2. Materials and methods

### 2.1. Ethics statement

The present study was conducted under the approval of Institutional Animal Care and Use Committee of Northeast Agricultural University (Heilongjiang province, China) (SYXK (Hei) 2012 – 2067) in accordance with Laboratory Animal-Guideline for ethical review of animal welfare (GB/T 35892 – 2018, National Standards of the People's Republic of China). Mice are anesthetized with ketamine and then euthanized, with all efforts made to minimize suffering.

### 2.2. Animal experiments

Six-week-old male C57BL/6 mice were purchased from Changsheng Biotechnology Co. (Liaoning, China). Throughout the experimental period, the temperature in the animal laboratory was maintained at $25 \pm 2\%$, the humidity was set at 45–55%, and the animals were subjected to a 12-hour light-dark cycle. After one week of acclimatization, the mice were allocated randomly into four groups: control group (CG), RSP group, RSP + SSA group, and SSA group. The mice in the RSP group and RSP + SSA group were administered RSP (Yuanye, Shanghai) intraperitoneally at a dose of 0.5 mg/kg from day 1–8, and 0.25 mg/kg for the following week, to induce the depression model. The blank control and SSA groups were given the same amount of solvent injection. On days 1–15, the groups receiving RSP + SSA and SSA were orally dosed with SSA (Yuanye, Shanghai) at a concentration of 50 mg/kg. The blank control and RSP groups were administered the same volume of saline by oral gavage. The experimental protocol and dosing regimen are illustrated in Fig 1A. At the conclusion of the experiment, all mice were anesthetized using isoflurane and subsequently euthanized via cervical dislocation. Blood and tissue samples were collected for use in relevant experiments or stored at −80 °C.

### 2.3. Body weight changes

All mice were weighed on days 1, 7 and 14 after grouping and were given only water for the first 12 hours of body weight testing.

### 2.4. Behavioral tests

All behavioral tests in this study were recorded without the recorder being informed of the purpose of the tests.

**2.4.1. The forced swimming test (FST).** The forced swimming test was conducted according to a previously established methodology [26]. Prepare a cylindrical container that is 50 cm tall and 22.5 cm in diameter, by adding warm water at a temperature of 25 °C until the water level reaches 37 cm. During the initial 2-minute period when mice were first placed in

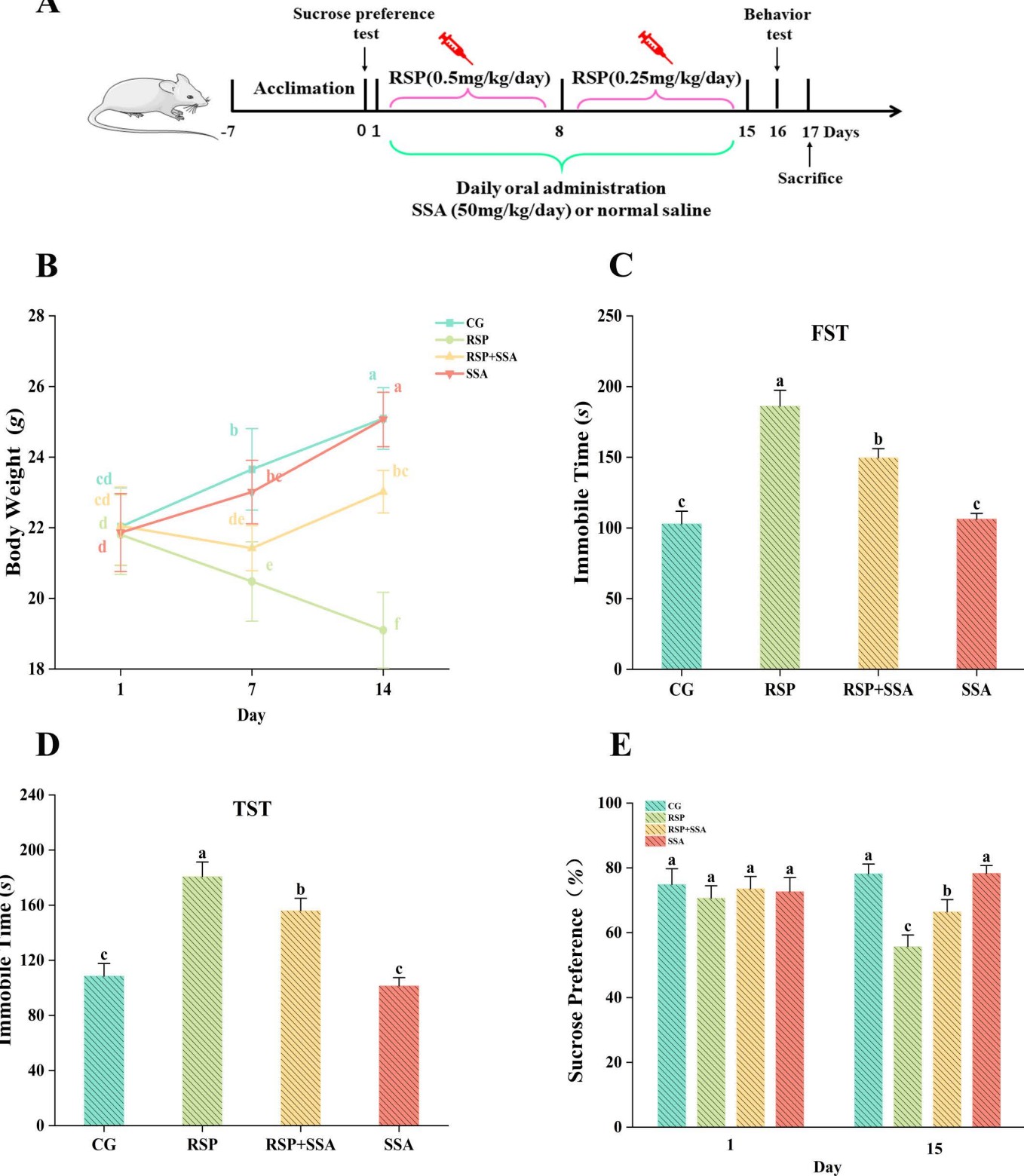

**Fig 1. Effects of SSA on the development and depressive-like behavior of RSP-induced depression in mice.** (A) Experimental flow chart, this chart was created through Microsoft Office PowerPoint and is in the free domain. (B) Changes in body weight of mice in different groups (n = 10). (C) Forced swimming test (n = 3). (D) The tail suspension test (n = 3). (E) Sucrose preference test (n = 3). Data are mean ± SD values. Bars with different lowercase letters indicate significant difference ($p < 0.05$).

the bucket, the duration of immobility was not recorded. Subsequently, over the following 4 minutes, the total duration of immobility was recorded when the mice reached a stable, typical immobile posture.

**2.4.2. The tail suspension test (TST).** The tail suspension test is conducted concerning the previously established methodology [27]. Briefly, mice were subjected to visual and acoustic isolation and restrained with tape approximately 1 cm from the tail tip, suspended 50 cm above the ground. Notably, only the last 4 minutes of the 6-minute immobility period were recorded for analysis.

**2.4.3. The sucrose preference tests (SPT).** Sucrose preference tests were carried out on day 1 and day 15 post-grouping. Prepare two bottles, one with water and the other with 1% sucrose solution, and place them on the cage at the same time. The mice underwent a 3-hour period of fasting and hydration before being allowed to access the bottles freely. The two bottles alternate positions every 8 hours. Sucrose preference was calculated using to the following formula: Sucrose preference = Sucrose intake (mL)/Total fluid intake (mL) x 100%.

## 2.5. Enzyme-linked immunosorbent assay (ELISA)

Determination of 5-HT, DA, NE, TNF-α, and IL-1β in mice was performed in strict accordance with the guidelines outlined by the manufacturer of the reagents (Nanjing JianCheng Bioengineering Institute, China). Out of these, 5-HT, DA and NE were measured with blood samples and the rest with homogenized hippocampal tissue.

## 2.6. Histopathological examination

Hematoxylin-eosin (HE) and Fluoro-Jade B (FJB) staining were executed according to previously defined techniques [28,29]. Briefly, the mouse hippocampi were removed meticulously on ice and submerged in 4% tissue fixative. This was then succeeded by dehydration, fixation, sectioning, and staining. Sections are observed through either a light or fluorescence microscope.

## 2.7. Detection of oxidant status

Measure the levels of lipid peroxidation (LPO) and total ROS in the hippocampus of the mouse according to the instructions provided by the reagent supplier. The hippocampus used for LPO and ROS assays was fresh mouse hippocampal tissue. Enzyme markers or inverted fluorescence microscopy were used to observe the results.

## 2.8. Real-time PCR (qPCR)

According to the instructions, total RNA was extracted from the mouse hippocampus using Trizol solution (Takala, Dalian, China), followed by the reverse transcription kit (Takala, Dalian, China) to transcribe the RNA into cDNA. Real-time qPCR reactions using a Roche Light Cycler instrument [30]. Results were analyzed with the $2^{-\Delta\Delta CT}$ method. With GAPDH as the internal reference, the sequences of the internal reference and the other primers are shown in detail in Table 1.

## 2.9. Western blot analysis (WB)

Western blotting assays refer to previous description [31]. In short, mouse hippocampal proteins were extracted from the RIPA lysate containing PMSF and their concentration was then measured using the BCA kit (Beyotime, China). The protein samples were treated with

**Table 1. List of primers used in qRT-PCR.**

| Gene | Primer sequence (5' – 3') | |
|---|---|---|
| GAPDH | Forward | TCGGGCCACGCTAATCTCAT |
| | Reverse | ACGGCCAAATCCGTTCACA |
| TNF-α | Forward | CCCAAAGGGATGAGAAGTTCC |
| | Reverse | GCTACAGGCTTGTCACTCGAA |
| IL-1β | Forward | TGCACCAGATGGATGACCAACTGCTTAGC |
| | Reverse | GGCATGGAGCGTTATTGCAACTGTGGTCATGAG |
| TNFR1 | Forward | AAATAGTCCTTCCTACCCCAA |
| | Reverse | CCGAGTAGATCTCAAAGTGAC |
| NF-κB | Forward | TGCCAAGAGTGATGACGAGGAGAG |
| | Reverse | TGA-GCGTGGAGGTGGATGATGG |

SDS buffer and heated in a metal bath at 95 °C for 10 minutes to denature them. The proteins were separated by a 12.5% separating gel and transferred to nitrocellulose membranes. Next, skimmed milk blocking, TBST wash, primary antibody treatment, secondary antibody treatment. Observation of the expression of proteins after treatment of membranes with enhanced chemiluminescence reagents.

## 2.10. 16S rRNA amplicon sequencing, data processing and analysis

DNA was extracted from mouse feces using the MagPure Stool DNA KF Kit (Magen, China) according to the reagent manufacturer's instructions. The quantification of DNA was performed using Nanodrop and the quality of the extract was checked by 1.2% agarose gel electrophoresis. PCR amplification was carried out using Pfu high-fidelity DNA polymerase (TransGen Biotech, China) in accordance with the protocol supplied by the reagent manufacturer. Amplified products were purified by adding a specific amount of Vazyme VAHT-STM DNA Clean Beads and were then eluted using an Elution Buffer. The DNA libraries were created employing the TruSeq Nano DNA LT Library Prep Kit and quantified using the Quant-iT PicoGreen dsDNA Assay Kit. Initial screening yielded raw sequence data, which were later classified into libraries and samples, based on Barcode and Index information. The barcode sequences were eliminated from the final samples. Finally, the sequences underwent quality monitoring and sequence splicing as per the Vsearch software. Bioinformatics analysis of the data was conducted via the personal bio-Gene Cloud platform (https://www.genes-cloud.cn/).

## 2.11. Statistical analysis

All experiments were carried out at least three times (n = 3), and the simultaneous values are presented as mean ± standard deviation. One-way ANOVA with Duncan's multiple-range test was used to analyze statistically significant differences between these experimental groups using the Statistical Package for Social Sciences (SPSS, version 24.0).

## 3. Results

## 3.1. Effects of SSA on RSP-induced weight changes and behavioral tests

The weights of the mice were monitored on a weekly basis. Fig 1B indicates that no significant differences were observed within any of the groups on the initial day. Persistent weight loss was observed in mice from the RSP group following the reserpine challenge. However, the

oral administration of SSA inhibited weight loss and, compared with the RSP group, the RSP + SSA group improved the RSP injection-induced weight loss trend to some extent by day 7 and reversed the downward trend with weight gain by day 14 ($p < 0.05$)

In this investigation, FST, TST and SPT were employed to assess anhedonia, despondent behavior, and the therapy of these symptoms by SSA in mice. In the FST and TST tests, immobility time was significantly prolonged in RSP-treated mice compared to the CG group, whereas it was shortened after SSA treatment (Fig 1C and D). In the initial SPS tests, all groups of mice displayed a liking for sugar water (Fig 1E). In the subsequent trial, some changes were observed. Specifically, mice in the RSP group increased their consumption of pure water, whilst the preference for sucrose water significantly reduced, in comparison to the RSP + SSA group ($p < 0.05$). In both the weight change and behavioral test trials, there was no discernible statistical difference between the SSA and CG groups ($p > 0.05$). The findings indicate that RSP can cause the emergence of depressive-like behaviors in mice, and that SSA treatment may effectively in stalling the exacerbation of these symptoms.

### 3.2. Effect of SSA on monoamine neurotransmitters in RSP-induced mice

Levels of monoamine neurotransmitters in the body are an important indicator of mood disorders. As illustrated in Fig 2, RSP-induced depressed mice showed a remarkable decrease in blood monoamine neurotransmitters compared to the CG group. Oral administration of SSA was effective in improving DA levels in the blood of the mice ($p < 0.05$), but they were still lower than in the CG group ($p < 0.05$). In addition, the modulation of NE levels by SSA showed no significant effects. Surprisingly, SSA-treated 5-HT levels were significantly elevated, with no significant difference compared to the CG group ($p > 0.05$). Oral administration of SSA alone does not significantly alter monoamine neurotransmitter levels, suggesting that the modulation of these neurotransmitters by SSA is not a direct mechanism of its antidepressant action. However, our results indicate that SSA treatment may partially counteract the RSP-induced reduction in monoamine neurotransmitter levels, offering a potential therapeutic strategy for depression.

### 3.3. Effects of SSA on RSP-induced damage to the hippocampus

Hippocampal neuronal damage and reduction is a marker of clinical depression. Results from HE showed that RSP treatment resulted in fewer neurons and structural abnormalities in the mouse hippocampus (Fig 3C). FJB staining is utilized to ascertain neuronal degeneration. In this experiment, neuronal cell death in the hippocampus was detected using FJB staining (Fig 3A). The findings indicate a significant increase in the number of positive cells within the hippocampus of mice in the RSP group when compared to the CG group. However, this effect was mitigated effectively in the RSP + SSA group (Fig 3A and B). This indicates that the administration of SSA hinders the degeneration of neurons induced by RSP in the hippocampus.

### 3.4. Effect of SSA on RSP-induced oxidative stress in the hippocampus

Oxidative stress is a negative impact caused in the body by free radicals that is regarded to be a major factor in hippocampus damage. In this study, we utilized LPO and ROS to evaluate the oxidative stress status of the hippocampus. The levels of ROS in the hippocampus were appreciably higher in the mice that received RSP injections in comparison to those in the CG group ($p < 0.05$), but the hippocampal ROS levels were appreciably lower in the SSA-treated mice compared to those in the RSP group ($p < 0.05$). The LPO results suggest that SSA regulated RSP-induced oxidative stress effectively, thereby controlling the LPO concentration in the hippocampus of

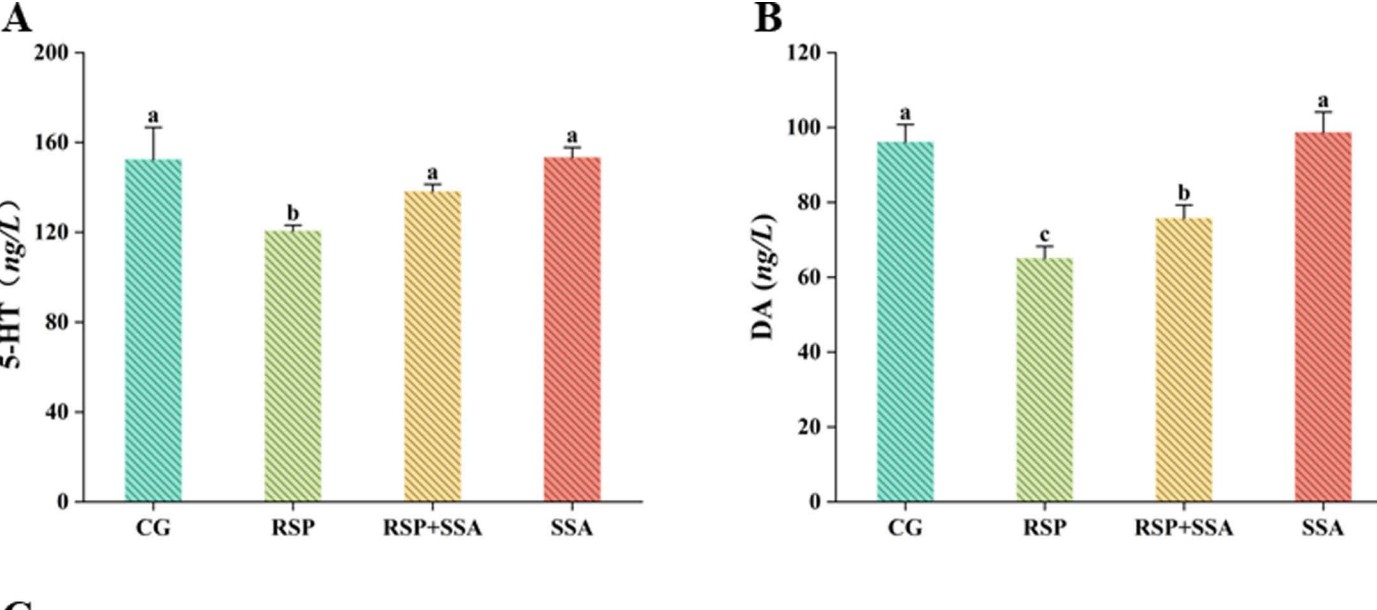

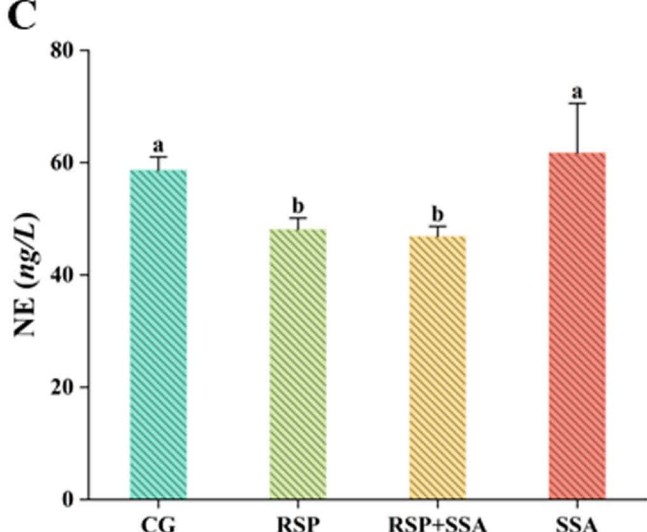

**Fig 2. Effect of SSA on monoamine neurotransmitters in RSP-induced depression in mice.** Determination of monoamine neurotransmitter content by ELISA for (A) 5-hydroxytryptamine (B) Dopamine and (C) Noradrenaline. Data are mean ± SD values (n = 3). Bars with different lowercase letters indicate significant difference ($p < 0.05$).

mice in the RSP + SSA group as compared to the RSP group ($p < 0.05$). The data indicates that SSA plays an antioxidant role in mitigating RSP-induced hippocampal damage (Fig 4).

### 3.5. Effect of SSA on RSP-induced pro-inflammatory factors in the hippocampus of mice

The presence of pro-inflammatory factors such as IL-1β and TNF-α in the brain damages neurons, causing neuroinflammation. As shown in Fig 5C and D, the levels of IL-1β and TNF-α mRNA were markedly increased in the hippocampus of mice in the RSP group compared with the CG group ($p < 0.05$), and there were no significant changes in the levels of IL-1β and TNF-α in the SSA group ($p > 0.05$). Notably, while the mRNA level of IL-1β in the

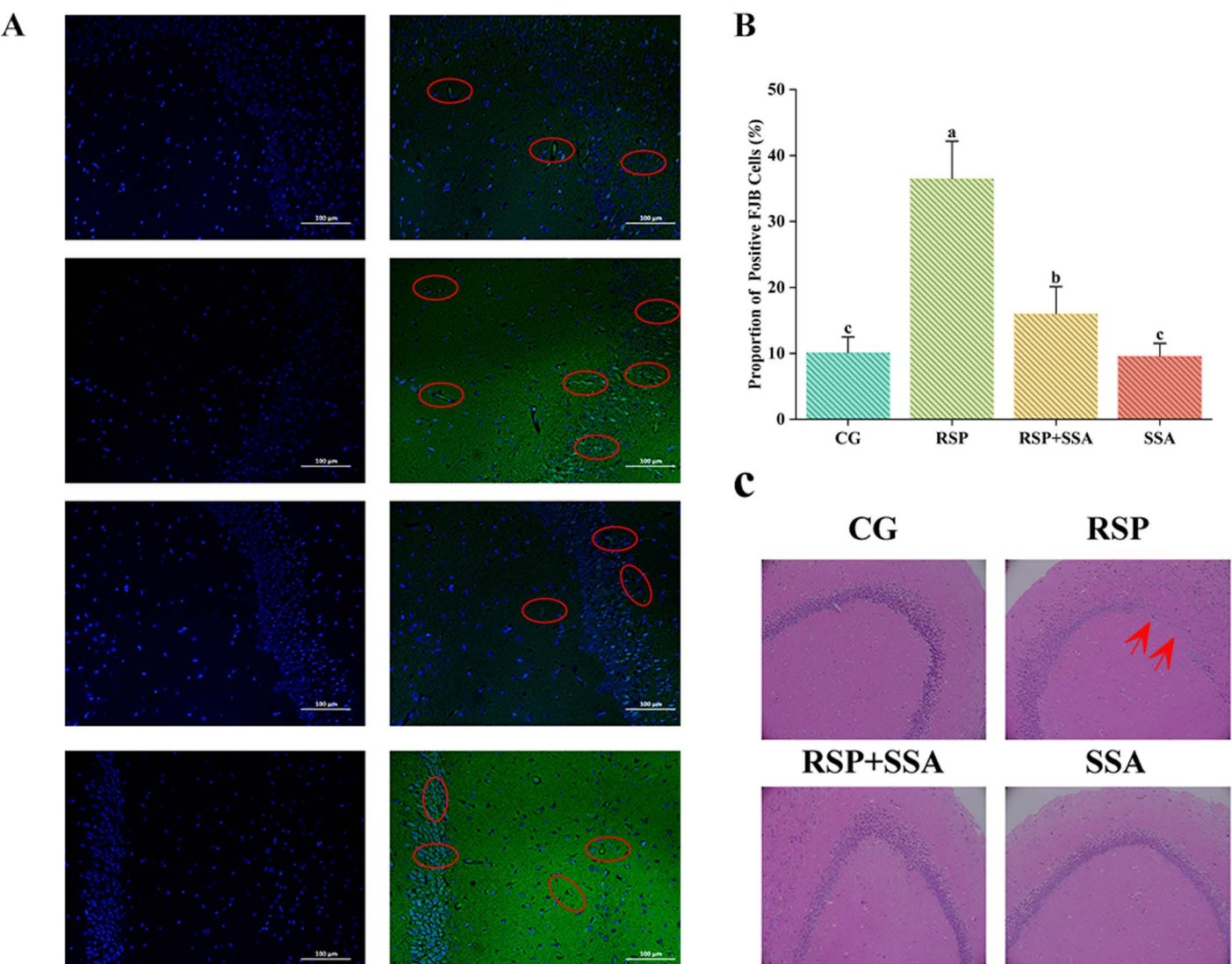

**Fig 3. Effects of SSA on RSP-induced damage to the hippocampus.** (A) FJB staining results, Cell nuclei appear blue, while neuronal cells undergoing degeneration are displayed in green, red circle indicate FJB-positive cells. (B) Proportion of FJB-positive cells in hippocampus (n = 3). (C) HE staining results (n = 3), red arrows indicate reduced neuronal cells in the hippocampus with structural abnormalities.

RSP + SSA group remained significantly higher than that of the CG group ($p < 0.05$), TNF-α had decreased to the same level as the CG group ($p > 0.05$). The ELISA findings were in agreement with the above results that intraperitoneal injection of RSP results in the elevation of pro-inflammatory factors in the hippocampus, however, IL-1β and TNF-α levels in the hippocampus of mice after SSA treatment were significantly regulated, especially TNF-α. These results suggest that SSA is involved in the immune response of the nervous system.

### 3.6. Effect of SSA on RSP-induced activation of TNFR1/NF-κB signaling pathway in mouse hippocampus

To explore the mechanism of action of SSA in reducing pro-inflammatory factors, we investigated the effect of SSA on the TNFR1/NF-κB pathway. As displayed in Figs 5A, B and 6, the

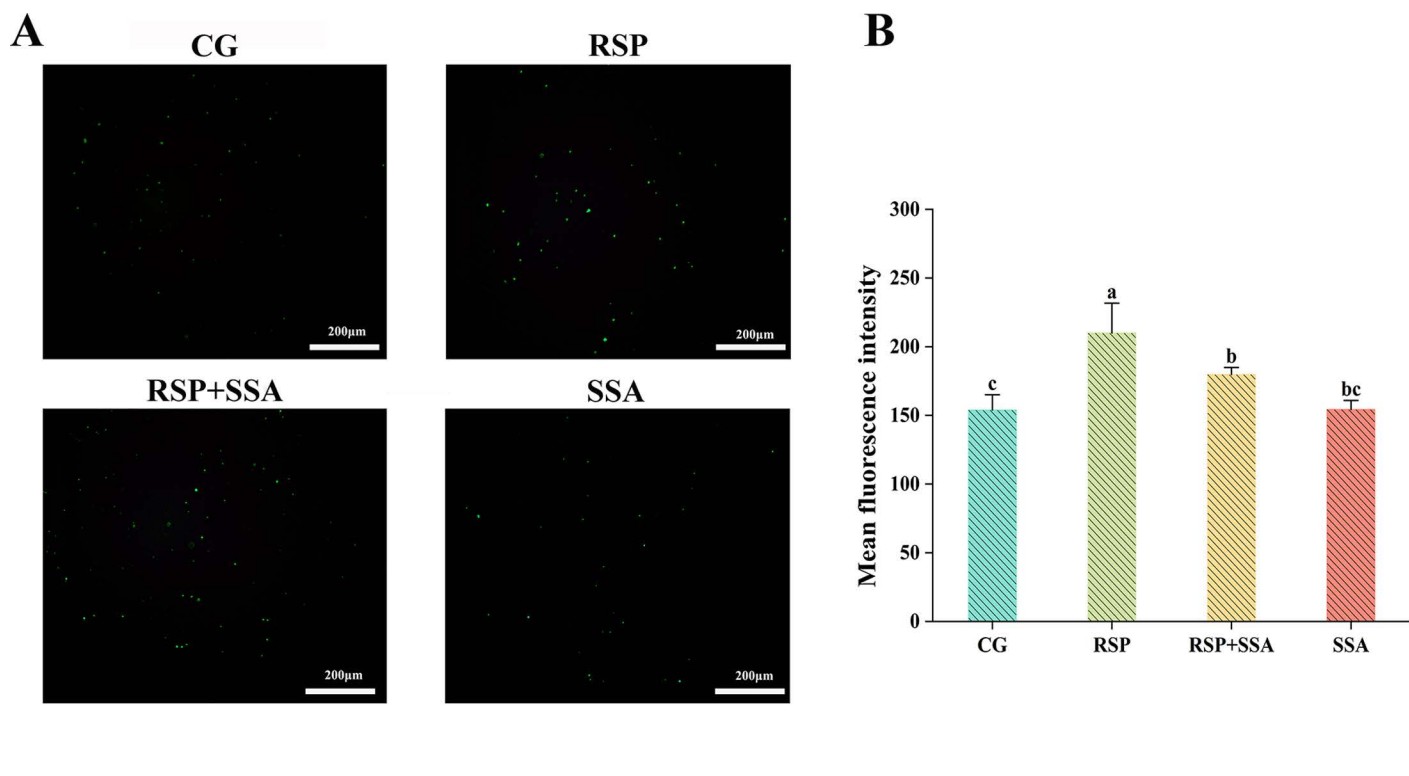

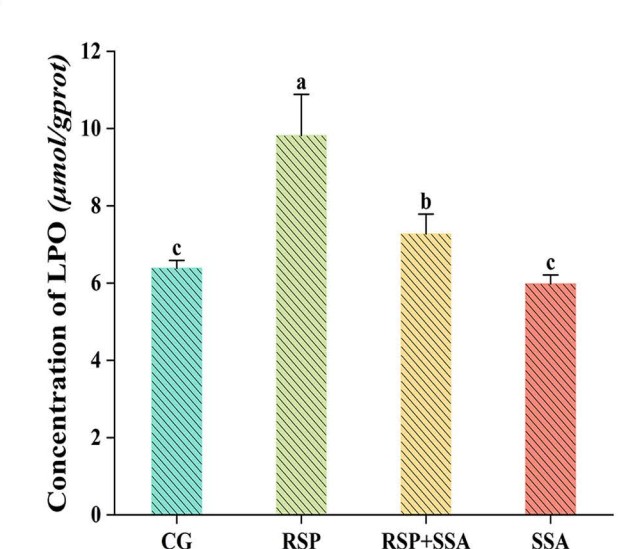

**Fig 4. Effect of SSA on RSP-induced oxidative stress in mouse hippocampus.** (A) The ROS Fluorescence staining graphs of each group of hippocampal cells under a light microscope (40×). (B) Analysis of DCFH-DA Fluorescent Staining Results. (C) The concentration of lipid peroxides in the mouse hippocampus. Data are mean ± SD values (n = 3). Bars with different lowercase letters indicate significant difference ($p < 0.05$).

mRNA levels of TNFR1 and NF-κB were significantly increased in the RSP group compared with the CG group ($p < 0.05$), while the WB results showed that RSP lead to the overexpression of TNFR1 and phosphorylated IκB proteins in the hippocampus of mice ($p < 0.05$). In contrast, SSA treatment inhibited the TNFR1/NF-κB signaling pathway resulting in a notable reduction

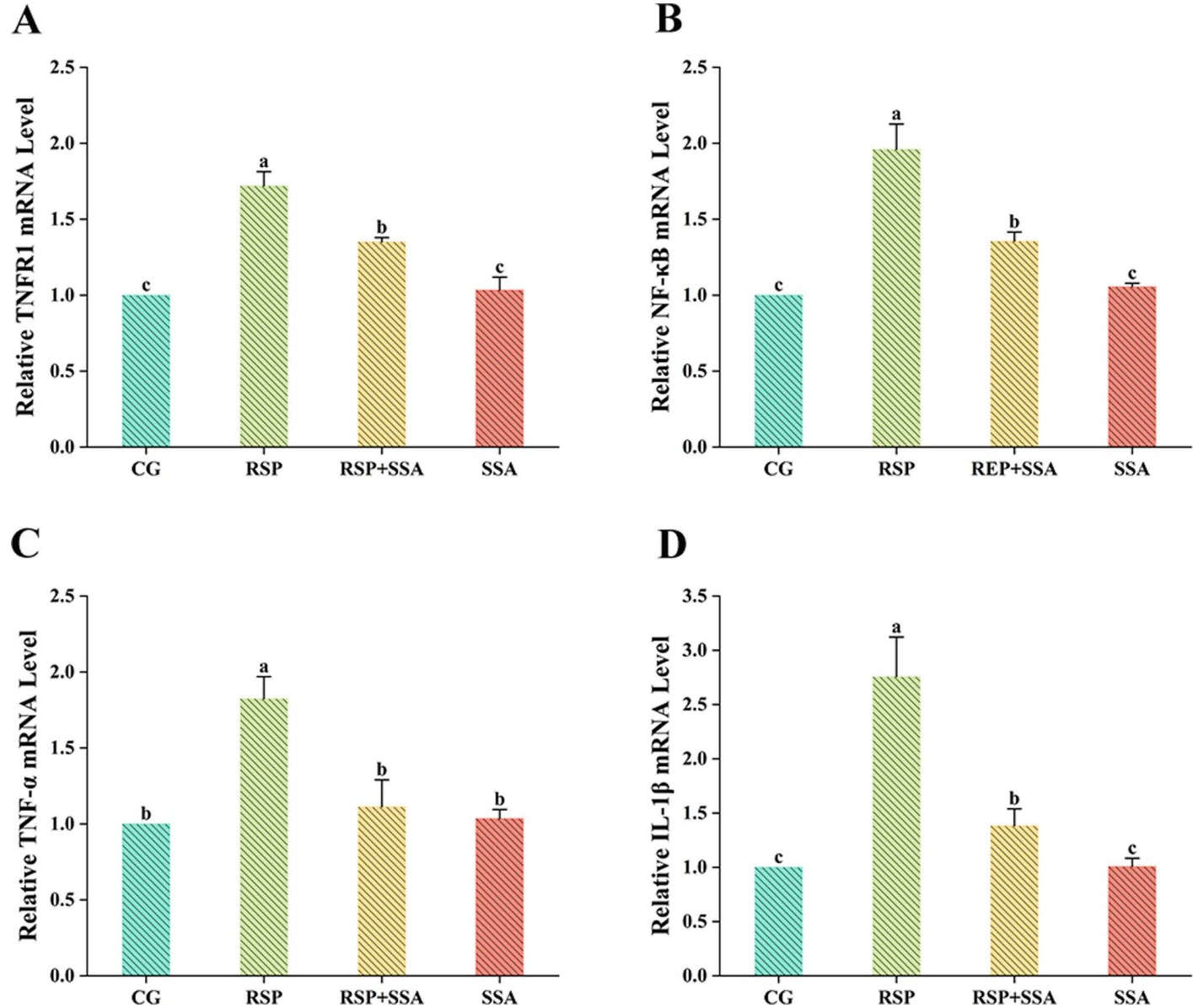

**Fig 5. Effect of SSA on mRNA expression of TNFR1/NF-κB signaling pathway-related indicators in the hippocampus of RSP-induced depressed mice.** Determination of mRNA expression levels of (A) TNFR1 (B) NF-κB (C) TNF-α and (D) IL-1β by qPCR. Data are mean ± SD values (n = 3). Bars with different lowercase letters indicate significant difference ($p < 0.05$).

in related mRNA and protein expression levels. Meanwhile, there was no significant difference in the mRNA and protein expression levels of TNFR1 and NF-κB in the SSA group compared to the CG group. The findings demonstrate that SSA treatment suppresses the activation of the TNFR1/NF-κB pathway in the hippocampus of mice in the RSP-induced depression model.

## 3.7. Effect of SSA on the RSP-induced intestinal flora of mice

Previous research indicates that modifying the gut microbiota could be a promising intervention for managing or alleviating depressive symptoms. In this experiment, the overall configuration of the gut microbiome in mouse fecal matter was analyzed by sequencing the

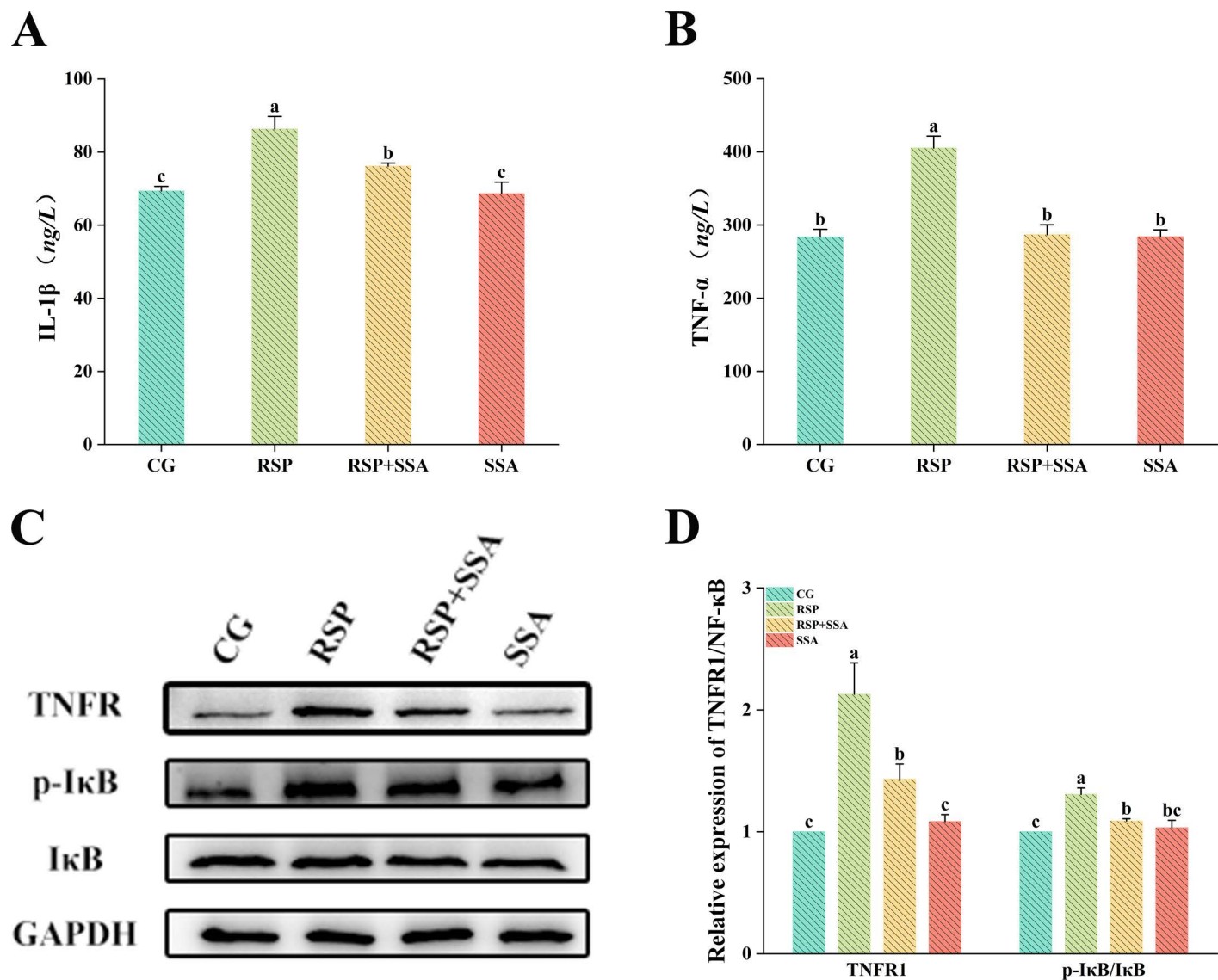

**Fig 6. Effects of SSA on TNFR1/NF- κB protein expression and TNF-α and IL-1β concentrations in the hippocampus of RSP-induced depressed mice.** Determination of inflammatory factors (A) IL-1β and (B) TNF-α in the hippocampus by ELISA. The expression of proteins related to the TNFR1/NF-κB signaling pathway in mouse hippocampus was demonstrated in (C) and (D). Data are mean ± SD values (n = 3). Bars with different lowercase letters indicate significant difference ($p < 0.05$).

16S rRNA gene. As shown in Fig 7A and B, the Chao1 index and Shannon index showed that the dilution curves of all groups increased sharply and then flattened as the sequencing depth increased, indicating that the 16S rRNA sequencing results of the contents of the mouse small intestine met the expected depth requirements and could be analyzed in the next step. At the gate level, there was no significant variation in the relative abundance of RSP in *Firmicutes*, *Verrucomicrobia*, and *Actinobacteria* when compared to the CG group, except for *Bacteroidetes*. However, in SSA-treated mice, a different outcome emerged, with *Firmicutes* being down-regulated and *Verrucomicrobia* being up-regulated in RSP + SSA as opposed to SSA (Fig 7C–E). The results of Alpha and Bete analyses showed that there was a significant separation of the gut microbiota between the four groups of mice, in addition, the RSP group

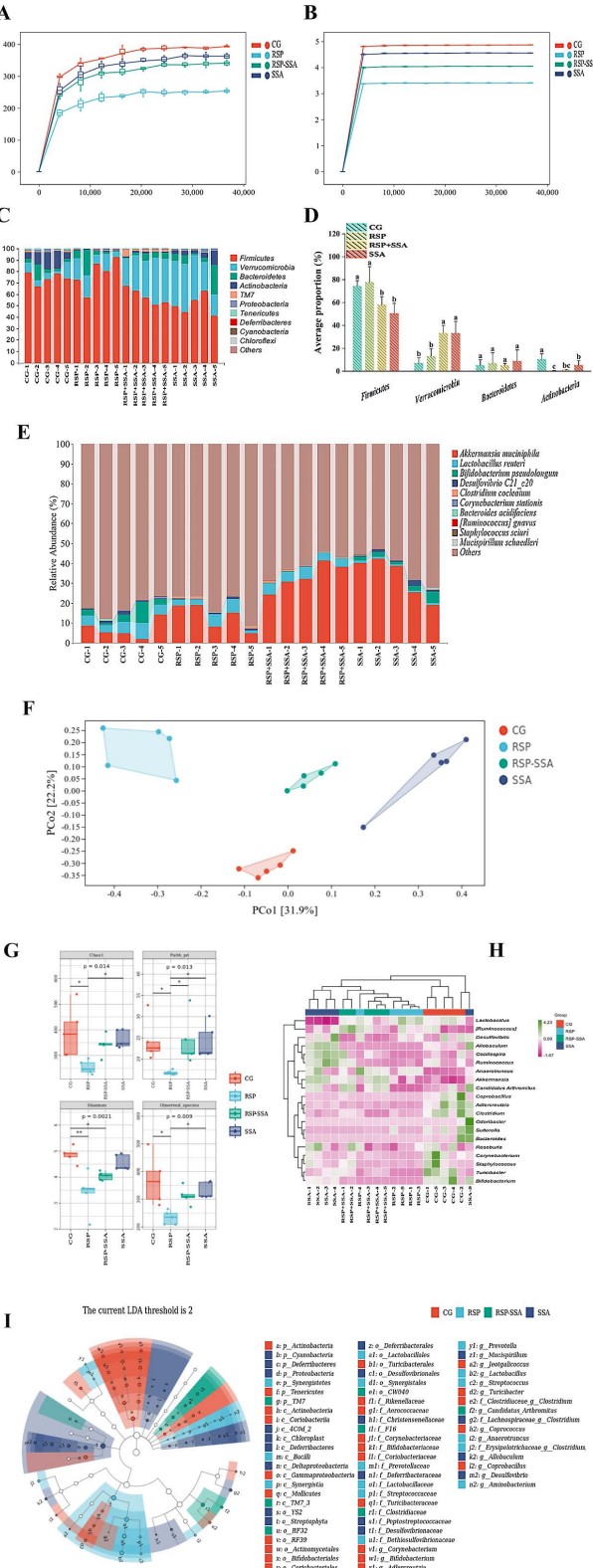

**Fig 7. Effect of SSA on the gut flora of RSP-induced depressed mice.** Rarefaction curves of the gut microbiota in each group were analyzed using (A) Chaos and (B) Shannon indices. Taxonomic analyses of bacteria at the phylum and genus levels in various gut microbiota groups are represented by (C–E). (F) NMSD analysis between different

groups. (G) Analysis of alpha diversity between different groups. (H) Thermograms were analyzed between different groups. (I) Analysis of LefSe between different groups. Data are mean ± SD values (n = 5). Bars with different lower-case letters indicate significant difference ($p < 0.05$).

showed a significant decrease in Chao1, Faith-pd, Shannon and Observed-species indices (Fig 7F and G). As depicted in Fig 7H, the thermogram exhibits that at the genus level, there exists a certain degree of independence in the composition of the gut microbiota of the four groups of mice. The results of the LefSe analysis indicated *Actinobacteria* and *Tenericutes* as the dominant bacteria in the gut microbiota of mice in the CG group. Whereas, *Synergistetes* were found to be dominant in the RSP group, and *TM7* in the RSP + SSA group. Moreover, *Proteobacteria*, *Cyanobacteri*, and *Deferribacteres* were identified as the dominant bacteria in the SSA group (Fig 7I). These findings demonstrate that SSA significantly modulated the gut microbiota in RSP-challenged mice, thereby inducing novel and significant shifts in the composition of the pre-existing gut microbiota.

## 4. Discussion

Currently, there is insufficient evidence to specify the precise pathogenesis of depression according to scientific research [17]. However, several drugs have been used by researchers to treat various animal models of depression to support their hypothesis [32]. SSA is the active ingredient in Chaihu-Shugan-San, which also has a potential role in the treatment of depression. Current evidence suggests that the antidepressant mechanism of SSA is in accordance with the hippocampal neuronal regeneration disorder hypothesis, whereby enhancement of the state of hippocampal neurons alleviates depressive-like behaviors in animals [22,23,33]. Furthermore, a study has illustrated that SSA may serve as a viable therapeutic option for severe acute pancreatitis (SAP). The intervention of SSA reduced the inflammatory response and oxidative signaling by altering the abundance of intestinal flora in rats [34]. This indicates that there are unexplored mechanisms for treating depression with SSA, particularly in addition to the hippocampal neural deficit hypothesis. Induction using RSP is a viable method of modeling depression and can effectively assess the effects of antidepressants [35]. Here, we investigated the therapeutic effects of SSA in a model of RSP-induced depression. The results showed that SSA could exert antidepressant effects by increasing blood monoamine levels, attenuating hippocampal inflammatory damage and oxidative stress, and altering gut microbiota in mice.

There is strong evidence from numerous studies that a dynamic homeostasis between the gut flora and the host has been shown to exist [36]. Once this balance is upset, the flora can impact the host's health via many pathways, such as energy absorption, endotoxemia and the cerebral-intestinal axis [37]. Our findings demonstrated that the RSP and SSA treatments changed the intestinal flora of the mice. Notably, in contrast to the CG and RSP groups, the SSA treatment increased the relative abundance of *Verrucomicrobia* while decreasing the relative abundance of *Firmicutes*. Aika Kosuge and colleagues discovered an increased abundance of *Verrucomicrobia* in the gut microbiome of a mouse model with depression induced by chronic social stress, and that reducing *Verrucomicrobia* led to improved depression symptoms [38]. However, there is some evidence that *Verrucomicrobia* abundance is negatively associated with inflammatory bowel disease, hypertension, depression, and premature aging [39–41]. Therefore, the mechanisms by which *Verrucomicrobia* plays a role in different diseases still need to be explored. Here, our results support the beneficial properties of *Verrucomicrobia*, at least in this study. The study revealed that in RSP-challenged mice, the gut microbiota exhibited a diminished abundance of *Verrucomicrobia*. In contrast, mice

administered with SSA displayed a significant increase in *Verrucomicrobia* levels, despite RSP exposure. Furthermore, a detailed analysis of *Verrucomicrobia* species indicated a substantial enrichment of *Akkermansia muciniphila* in response to SSA treatment. Since *Akkermansia muciniphila* was discovered and identified in the early 20th century, a large body of literature has shown that its deficiency or reduced abundance is associated with a variety of diseases, including diabetes, obesity, inflammation and depression, and it has been hailed as a paradigm for the next generation of beneficial microbes [42]. Recent research indicates that the provision of *Akkermansia muciniphila* supplements facilitates the metabolism of host β-alanyl-3-methyl-lhistidine and edaravone, and that the administering of these two metabolites in isolation has a marked effect on the treatment of chronic stress-induced depressive symptoms [43]. Interestingly, it was found that Amuc_1100, an outer membrane protein from *Akkermansia muciniphila*, could ameliorate chronic unpredictable mild stress-induced anxiety and depression-like behavior in mice [44]. Another study proposes that *Akkermansia muciniphila* improves symptoms similar to depression in mice by repairing the intestinal barrier and reducing intestinal inflammation [45]. Furthermore, the SSA treatment lowered the abundance of *Firmicutes* as well as the *Firmicutes/Bacteroidetes* ratio, which is beneficial for controlling host inflammation [46,47]. Taken together, treatment with SSA alleviates symptoms of depression in mice induced by RSP by altering the composition of the gut microbiota, however, the exact mechanism requires further investigation. It should be noted that the gut flora of humans and mice differ between species, and the effect of SSA on the human gut flora needs to be further investigated.

In recent years, research has increasingly clarified the role of the gut microbiome in the brain-gut axis. The gut microflora is implicated in host metabolism, immune response, and even monoamine neurotransmitter synthesis [12]. The monoamine hypothesis indicates a close connection between depression and monoamine neurotransmitters [48]. However, RSP treatment disrupts the function of vesicular monoamine transporter 2 (VMAT2), thereby inhibiting the reuptake of monoamine neurotransmitters. This inhibition leads to an excessive accumulation of these neurotransmitters in the synaptic cleft. Contrary to expectations, this overaccumulation does not induce hyperexcitability of neural activity. On the contrary, it enhances the exposure of monoamines to monoamine oxidase (MAO) within the synaptic cleft, promoting oxidative catabolism. This process suppresses excitatory stimuli in the brain, potentially triggering the expression of depressive-like behaviors. Common behavioral assays for assessing depression include the FST, TST and SPT. The current experiment utilized immobility time in mice and preference for sucrose water to confirm the establishment of the depression model. The findings revealed a considerable increase in immobilization time and a decrease in desire for sucrose water in the RSP group compared to the CG group, behaviors that were assumed to indicate the mice's despair and lack of happiness. However, treatment with SSA effectively improved exercise duration, reduced immobility time, increased sucrose water consumption, and inhibited RSP-induced weight loss in mice. Interestingly, treatment with SSA also reversed RSP-induced decreases in 5-HT and DA. These data appear to indicate that SSA treatment may have a beneficial effect on depression-like symptoms in mice by increasing levels of monoamine neurotransmitters. In fact, key players in vesicle fusion and release processes include SNARE and Syt1, proteins associated with the release of neurotransmitters. SNARE proteins are distributed across synaptic vesicles and presynaptic membranes and are involved in the core fusion mechanism [49]. In summary, it is conjectured that that SSA not only plays a role in regulating the gut microbiota, but also affects monoamine transmitter levels in mice, including the regulation of SNARE, Syt1 and $Ca^{2+}$. However, the mechanism by which the microbiota affects monoamine neurotransmitter levels remains to be investigated.

Neuropathophysiology demonstrates that nerve cell damage and atrophy in the hippocampus serve as structural indicators for the onset of depression. In this experiment, the results of histopathological sections showed that RSP-treated hippocampus can reduce the number of neurons, structural abnormalities and cause neuronal degeneration. Furthermore, elevated levels of pro-inflammatory factors IL1-β and TNF-α were observed in the hippocampus, accompanied by an upregulation in LPO concentration and ROS levels. In agreement with previous studies, SSA possessed anti-inflammatory and antioxidant activities [22] that significantly inhibited inflammation and oxidative stress in RSP-challenged mice. As research into inflammation and depression deepens, the cytokine doctrine of depression is becoming more acknowledged. Current evidence indicates that cytokine transduction can initiate numerous biological effects downstream, such as neuroendocrine, oxidative stress systems, and monoaminergic pathways. Subsequent changes in these systems can result in the 'disease behavior [50]. For a long time, TNF-α has been one of the molecules at the core of the study of the cytokine hypothesis of depression, and it has been consistently reported to be associated with the pathogenesis of depression [51]. A study indicates that TNF-α exerts an impact on the levels of central tryptophan neurotransmitters by modulating glial cell 5-HT receptor activity [52]. This suggests that TNF-α may be implicated in the regulation of monoamine neurotransmitter homeostasis, which is implicated in depressive-like behavior in mice. During the trial, it was noted that TNF-α levels were restored to control levels after SSA treatment. Further investigation was conducted to understand the mechanism of action of SSA on TNF-α, specifically examining the impact of SSA on the TNFR1/NF-κB signaling pathway. The activation of the TNFR1 receptor, which contains a death domain, seems to promote cell death and inflammation. In contrast, TNFR2 receptor activation benefits cell survival and tissue regeneration [53]. Moreover, in neurons, an increase in TNF-α levels can rapidly boost excitatory synaptic strength through TNFR1, cause excessive $Ca^{2+}$ influx, leading to excitotoxicity, and ultimately foster neuronal apoptosis [54]. Overall, our findings indicate that SSA has the potential to lessen the release of pro-inflammatory markers, with a particular focus on TNF-α, by inhibiting the activation of the TNFR1/NF-κB signaling pathway, thereby ameliorating RSP-induced neuronal inflammatory injury in the mouse hippocampus. Based on the above results, it can be inferred that SSA may have multi-target effects or can affect changes in monoamine substances and inflammatory factors by mediating gut microbiota, but the specific mechanism needs further research.

In conclusion, this study explores three prevailing hypotheses regarding the pathogenesis of depression: the monoamine hypothesis, the gut-brain axis, and the role of inflammatory factors. Our study shows that SSA effectively ameliorates RSP-induced depressive-like symptoms in mice through several aspects involving monoaminergic transmitters, inflammation, oxidative stress and gut microbiota. Furthermore, our study results not only extend the understanding of SSA for the treatment of RSP-induced depression, but also provide data to support the clinical application of SSA.

## Supporting information

**S1 Data. All raw data in this study were uploaded in the form of supplementary materials, named as Raw data 1 and Raw data 2-1 and Raw data 2-2.**
(ZIP)

## Acknowledgments

We are grateful to Dr. Zhiyong Wu, Laboratory of Pharmacology and Toxicology, College of Animal Medicine, Northeast Agricultural University, for the support of the experiment.

## Author contributions

**Conceptualization:** Menglin Wang, Haojun Li, Xianwu Zhang, Zhaohui Yin.

**Data curation:** Haojun Li, Wenjing Zhang, Lu Jia.

**Formal analysis:** Haojun Li, Wenjing Zhang, Lu Jia.

**Funding acquisition:** Zhaohui Yin.

**Methodology:** Haojun Li, Li Zhang.

**Project administration:** Li Zhang, Yu Zhang.

**Software:** Menglin Wang, Wenjing Zhang, Miao Jia.

**Supervision:** Yu Zhang, Xianwu Zhang.

**Validation:** Menglin Wang, Shun Wang, Haifei Gao.

**Visualization:** Shun Wang, Miao Jia.

**Writing – original draft:** Menglin Wang, Haojun Li, Zhaohui Yin.

**Writing – review & editing:** Menglin Wang, Haojun Li, Xianwu Zhang, Zhaohui Yin.

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
