## [Decision Letter · Decision Letter 0]

17 Jul 2024

PONE-D-24-20060Saikosaponin A alleviates depressive-like behaviour induced by reserpine in mice by regulating gut microflora and inflammatory responsesPLOS ONE

Dear Dr. Yin,

Thank you for submitting your manuscript to PLOS ONE. After careful consideration, we feel that it has merit but does not fully meet PLOS ONE’s publication criteria as it currently stands. Therefore, we invite you to submit a revised version of the manuscript that addresses the points raised during the review process.

We look forward to receiving your revised manuscript.

Kind regards,

Peng Zhong, Ph.D.

Academic Editor

PLOS ONE

Journal Requirements:

2. PLOS requires an ORCID iD for the corresponding author in Editorial Manager on papers submitted after December 6th, 2016. Please ensure that you have an ORCID iD and that it is validated in Editorial Manager. To do this, go to ‘Update my Information’ (in the upper left-hand corner of the main menu), and click on the Fetch/Validate link next to the ORCID field. This will take you to the ORCID site and allow you to create a new iD or authenticate a pre-existing iD in Editorial Manager. Please see the following video for instructions on linking an ORCID iD to your Editorial Manager account: https://www.youtube.com/watch?v=_xcclfuvtxQ"

3. To comply with PLOS ONE submissions requirements, in your Methods section, please provide additional information regarding the experiments involving animals and ensure you have included details on (1) methods of sacrifice, (2) methods of anesthesia and/or analgesia, and (3) efforts to alleviate suffering

4. Thank you for stating the following financial disclosure: "This work was supported by the the Natural Science Foundation of Inner Mongolia (2021M508068) and High level Clinical Specialist Construction Project of Public Hospitals in the Capital Region of Inner Mongolia Autonomous Region (2023SGGZ0005)." 

5. Thank you for stating the following in the Acknowledgments Section of your manuscript: "This work was supported by the the Natural Science Foundation of Inner Mongolia (2021M508068) and High level Clinical Specialist Construction Project of Public Hospitals in the Capital Region of Inner Mongolia Autonomous Region (2023SGGZ0005)."

Please remove any funding-related text from the manuscript and let us know how you would like to update your Funding Statement. Currently, your Funding Statement reads as follows: "This work was supported by the the Natural Science Foundation of Inner Mongolia (2021M508068) and High level Clinical Specialist Construction Project of Public Hospitals in the Capital Region of Inner Mongolia Autonomous Region (2023SGGZ0005)."

7.  Regarding blot/gel data: PLOS ONE now requires that submissions reporting blots or gels include original, uncropped blot/gel image data as a supplement or in a public repository. This is in addition to complying with our image preparation guidelines described at https://journals.plos.org/plosone/s/figures#loc-blot-and-gel-reporting-requirements. These requirements apply both to the main figures and to cropped blot/gel images included in Supporting Information. If the manuscript is positively reviewed, we will ask the authors to provide any missing raw image data for blot/gel results when they submit their first revision. As part of your review, please ensure that figures reporting blot or gel images comply with the journal’s image preparation guidelines and that the original data are provided following the journal’s request.  If you have any questions or concerns about blot/gel figures or data for this submission, please email us at plosone@plos.org before issuing a decision letter.

Reviewers' comments:

Reviewer's Responses to Questions

**Comments to the Author**

1. Is the manuscript technically sound, and do the data support the conclusions?

Reviewer #1: Partly

Reviewer #2: Yes

2. Has the statistical analysis been performed appropriately and rigorously? 

Reviewer #1: No

Reviewer #2: Yes

3. Have the authors made all data underlying the findings in their manuscript fully available?

Reviewer #1: No

Reviewer #2: Yes

4. Is the manuscript presented in an intelligible fashion and written in standard English?

Reviewer #1: No

Reviewer #2: Yes

5. Review Comments to the Author

Reviewer #1: 1. The overall English writing is not very standard; the author should revise it to make the English writing more standard.

2. The authors also write carelessly, with many typos in the manuscript, which could have been easily avoided if they had double-checked the manuscript before submitting it.

3. What is the sex of the mice? Are they all male or female? The authors should indicate this in the main text or materials and methods.

4. Line 68: “hydroxytryptamine(5-HT)” should be “hydroxytryptamine (5-HT)” with a space before the parenthesis. The same applies to the entire manuscript.

5. Line 221: “statis-tically” should be “statistically.”

6. Line 236: What does “CG” mean? Does it mean “control group”? The author should write the full name when using the abbreviation for the first time. The same applies to “HE,” “FJB,” etc.

7. Figure 1: What is the meaning of letters “a,” “b,” “c,” “d”? The authors write: “Bars with different lowercase letters indicate significant difference (p < 0.05).” But which groups were compared to get the significant differences? The authors should clearly write this in the figure legends or indicate it in the figures. The same applies to all the figures.

8. Line 258: Another section “3.2”? This should be section “3.3”, as there is already a “3.2.”

9. Line 259: Grammar/typo: “Hippocampal neuronal damage and reduction as a marker of clinical depression.” Maybe the authors mean “Hippocampal neuronal damage and reduction is a marker of clinical depression.”?

10. Line 266: Typo: “RSP+SSA group This indicates that” should be “RSP+SSA group. This indicates that.”

11. For Figure 3, the authors should quantify the number of positive cells in the hippocampus before claiming that “The findings indicate a significant increase in the number of positive cells within the hippocampus of mice in the RSP group when compared to the CG group” (Line 263-265). The same applies to Figure 3B. The author should quantify the HE staining results to show that there are “reduced neuronal cells in the hippocampus with structural abnormalities.” The same applies to Figure 4.

12. Figure 4A: The staining in Figure 4A is too faint to see. The author should quantify the signal to show the differences.

Reviewer #2: With this manuscript titled “Saikosaponin A alleviates depressive-like behaviour induced by reserpine in mice by regulating gut microflora and inflammatory responses” the authors found SSA effectively ameliorates RSP-induced depressive-like symptoms in mice through several aspects involving monoaminergic transmitters, inflammation, oxidative stress and gut microbiota.

The manuscript is pleasant to read. But I still have some concerns that need to be addressed.

Comments:

1) Since the current study was only done in male mice, author should discussion the limitation of sex difference effect of SSA.

2) Page 2, line 119. How many mice were used in the study? I noted some group have N=10 mice, but other groups have N=3 mice. Total mice?

3) It’s not clear to reader that Solvent is the vehicle for RSP or not.

4) Page 7, line 122, administration of RSP is ip or po?

5) Figure 1. Groups with N=3 mice are smaller N number for the behavior assay; it will be good include p and F value for each figure from One-way ANOVA test.

6) Any acute effect of SSA on Day 1 treatment compared to 14 days SSA treatment effect? Any dose-dependent manner of SSA in low, middle and high dose?

7) Any PK profile and half-life data of SSA?

8) It’s not clear to the reader that the behavior tests were conducted at dark or light phase. As the rodent is notational animals, the phenotype of behavior will be difference between dark and light phase. Authors should consider and discuss it.

6. PLOS authors have the option to publish the peer review history of their article (what does this mean? ). If published, this will include your full peer review and any attached files.

**Do you want your identity to be public for this peer review?** For information about this choice, including consent withdrawal, please see our Privacy Policy .

Reviewer #1: No

Reviewer #2: No

---

## [Author Response · Author response to Decision Letter 0]

29 Aug 2024

Dear Editors and Reviewers,

We would like to express our gratitude for your valuable feedback on our manuscript. Your insights are invaluable in helping us to enhance and refine the manuscript. In light of the comments you have highlighted, we conducted a discussion and subsequent revision of the manuscript. Here, we resubmit the revised manuscript " Saikosaponin A alleviates depressive-like behavior induced by reserpine in mice by regulating gut microflora and inflammatory responses", (NO. PONE-D-24-20060). I would like to extend my gratitude once more to the editors of the journal “PLOS ONE” and the invited reviewers for your invaluable time and effort in the review process of this manuscript.

In light of the valuable feedback provided by the esteemed editors and reviewers, the manuscript has been meticulously revised. The comments that we responded to are presented below. The comments made by editors or reviewers are indicated in red, while our proposed revisions are indicated in blue. It is our sincere hope that you will be gratified by the results of our modification programme. It is our sincere hope that you will be satisfied with the revised manuscript.

Editorial requirements

Dear Editors,

Thank you for your advice. Based on the link you provided, we have made changes to the manuscript. We hope that the revised manuscript meets the requirements of the journal.

2. PLOS requires an ORCID iD for the corresponding author in Editorial Manager on papers submitted after December 6th, 2016. Please ensure that you have an ORCID iD and that it is validated in Editorial Manager. To do this, go to ‘Update my Information’ (in the upper left-hand corner of the main menu), and click on the Fetch/Validate link next to the ORCID field. This will take you to the ORCID site and allow you to create a new iD or authenticate a pre-existing iD in Editorial Manager. Please see the following video for instructions on linking an ORCID iD to your Editorial Manager account: https://www.youtube.com/watch?v=_xcclfuvtxQ.

Dear Editors,

Thank you for your valuable advice. Upon your request, we have registered the ORCID iD. Sincere wishes.

3. To comply with PLOS ONE submissions requirements, in your Methods section, please provide additional information regarding the experiments involving animals and ensure you have included details on (1) methods of sacrifice, (2) methods of anesthesia and/or analgesia, and (3) efforts to alleviate suffering

Dear Editors,

Thanks for the friendly advice. This part of the information has been included in the latest manuscript to ensure that animal welfare is safeguarded. Thanks again for the advice.

4. Thank you for stating the following financial disclosure: "This work was supported by the the Natural Science Foundation of Inner Mongolia (2021M508068) and High level Clinical Specialist Construction Project of Public Hospitals in the Capital Region of Inner Mongolia Autonomous Region (2023SGGZ0005)." Please state what role the funders took in the study. If the funders had no role, please state: ""The funders had no role in study design, data collection and analysis, decision to publish, or preparation of the manuscript."" If this statement is not correct you must amend it as needed Please include this amended Role of Funder statement in your cover letter; we will change the online submission form on your behalf.

Dear Editors,

Thank you for your advice. In the cover letter we make a detailed presentation. We hope our answer satisfies you.

Author Contributions

Conceptualization: Xianwu Zhang, Zhaohui Yin and Menglin Wang.

Data curation: Menglin Wang, Haojun Li, Haifei Gao, Shun Wang, Yu Zhang.

Formal analysis: Menglin Wang, Wenjing Zhang, Miao Jia, Haifei Gao, Xianwu Zhang

Funding acquisition: Menglin Wang, Zhaohui Yin.

Investigation: Wenjing Zhang, Li Zhang, Miao Jia.

Methodology: Menglin Wang, Haojun Li, Shun Wang, Lu Jia, Yu Zhang.

Supervision: Haifei Gao, Yu Zhang.

Validation: Menglin Wang, Haojun Li, Li Zhang, Haifei Gao, Lu Jia.

Writing – original draft: Menglin Wang, Haojun Li, Li Zhang.

Writing – review & editing: Xianwu Zhang, Zhaohui Yin, Menglin Wang.

Both grant recipients participated in the experiment.

This work was supported by the Natural Science Foundation of Inner Mongolia (2024QN08067) and High level Clinical Specialist Construction Project of Public Hospitals in the Capital Region of Inner Mongolia Autonomous Region (2023SGGZ0005). Both grant recipients participated in the experiment.

5. Thank you for stating the following in the Acknowledgments Section of your manuscript: "This work was supported by the the Natural Science Foundation of Inner Mongolia (2021M508068) and High level Clinical Specialist Construction Project of Public Hospitals in the Capital Region of Inner Mongolia Autonomous Region (2023SGGZ0005)." We note that you have provided funding information that is not currently declared in your Funding Statement. However, funding information should not appear in the Acknowledgments section or other areas of your manuscript. We will only publish funding information present in the Funding Statement section of the online submission form. Please remove any funding-related text from the manuscript and let us know how you would like to update your Funding Statement. Currently, your Funding Statement reads as follows: "This work was supported by the the Natural Science Foundation of Inner Mongolia (2021M508068) and High level Clinical Specialist Construction Project of Public Hospitals in the Capital Region of Inner Mongolia Autonomous Region (2023SGGZ0005)."

Dear Editors,

Thank you for your advice. In response to your suggestion, we have removed this information from the acknowledgements section of the manuscript. Meanwhile we have updated our funding information.

This work was supported by the Natural Science Foundation of Inner Mongolia (2024QN08067) and High level Clinical Specialist Construction Project of Public Hospitals in the Capital Region of Inner Mongolia Autonomous Region (2023SGGZ0005).

Dear Editors,

Thank you for your advice. In the latest manuscript, we have made changes accordingly. Thanks again for the advice.

2.1 Ethics statement

The present study was conducted under the approval of Institutional Animal Care and Use Committee of Northeast Agricultural University (Heilongjiang province, China) (SYXK (Hei) 2012–2067) in accordance with Laboratory Animal-Guideline for ethical review of animal welfare (GB/T 35892–2018, National Standards of the People's Republic of China). Mice are anesthetized with ketamine and then euthanized, with all efforts made to minimize suffering.

7. Regarding blot/gel data: PLOS ONE now requires that submissions reporting blots or gels include original, uncropped blot/gel image data as a supplement or in a public repository. This is in addition to complying with our image preparation guidelines described at https://journals.plos.org/plosone/s/figures#loc-blot-and-gel-reporting-requirements. These requirements apply both to the main figures and to cropped blot/gel images included in Supporting Information. If the manuscript is positively reviewed, we will ask the authors to provide any missing raw image data for blot/gel results when they submit their first revision. As part of your review, please ensure that figures reporting blot or gel images comply with the journal’s image preparation guidelines and that the original data are provided following the journal’s request. If you have any questions or concerns about blot/gel figures or data for this submission, please email us at plosone@plos.org before issuing a decision letter.

Dear Editors,

Thank you for your advice. In the latest manuscript, our internal reference and the target protein are located on different membranes. Due to the proximity between the target protein and the internal reference, we conducted the experiments separately. The internal reference (GAPDH) and target proteins in our Western blot (WB) analyses are derived from the same batch of samples. To guarantee the precision of the experimental results, the WB experiments were conducted at least three times for each protein. We have uploaded the original Western Blot images of the respective experiment. We sincerely hope our response meets your satisfaction. Please accept this expression of sincerest gratitude for your invaluable counsel. Thanks again for the advice.

Reviewer #1:

1. The overall English writing is not very standard; the author should revise it to make the English writing more standard.

Dear Reviewers and Editors,

Thank you for your precious comments. In order to improve manuscript writing problems, we revised the manuscript to improve its readability. In addition, we invited the language editor to revise the language and style of the manuscript. We appreciate your meaningful input and sincerely hope our response meets with your approval.

2. The authors also write carelessly, with many typos in the manuscript, which could have been easily avoided if they had double-checked the manuscript before submitting it.

Dear Reviewers and Editors,

Thank you for taking the time to provide us with your feedback. We apologize for our carelessness. We have rechecked the manuscript several times and have corrected possible typos in the article. We hope our response meets with your approval. Thank you once more for your invaluable counsel.

3. What is the sex of the mice? Are they all male or female? The authors should indicate this in the main text or materials and methods.

Dear Reviewers and Editors,

We would like to express our gratitude for your valuable suggestions, which will undoubtedly contribute to the enhancement of the quality of our manuscripts. The sex of the mice (male) has been indicated in the manuscript Materials and Methods. We would like to thank you for your worthy suggestion.

4. Line 68: “hydroxytryptamine(5-HT)” should be “hydroxytryptamine (5-HT)” with a space before the parenthesis. The same applies to the entire manuscript.

Dear Reviewers and Editors,

The expression of gratitude is extended for the invaluable feedback provided. Such errors in the manuscript have been corrected in their entirety. We would like to express our gratitude once more for your endeavors to enhance the calibre of our manuscript.

5. 5. Line 221: “statis-tically” should be “statistically.”

Dear Reviewers and Editors,

We are grateful for your professional review work on our article. We've corrected the error. Please accept this expression of sincerest gratitude for your invaluable counsel.

6. Line 236: What does “CG” mean? Does it mean “control group”? The author should write the full name when using the abbreviation for the first time. The same applies to “HE,” “FJB,” etc.

Dear Reviewers and Editors,

Thank you for pointing out this issue in the manuscript. The abbreviation CG means control group. We've rechecked the manuscript to make sure all the abbreviations have full names. In addition, we have collated all the abbreviations in the manuscripts. Thank you once again.

SSA, Saikosaponin A; RSP, Reserpine; ROS, Reactive oxygen species; IL-1β, Interleukin-1beta; TNF-α, Tumor Necrosis Factor-α; 5-HT, 5-hydroxytryptamine; DA, Dopamine; NE, Norepinephrine; Bax, BCL2-Associated X; CREB, cAMP-response element binding protein; BDNF, Brain-derived neurotrophic factor; CG, Control group; TST, Tail Suspension Test; SPT, Sucrose Preference Test; ELISA, Enzyme-Linked Immunosorbent Assay; HE, Hematoxylin-eosin; FJB, Fluoro-Jade B; LPO, Lipid peroxidation; TNFR1, Tumor necrosis factor receptor-1; NF-κB, Nuclear Factor kappa-light-chain-enhancer of activated B cells; SAP, Severe acute pancreatitis; SNARE, soluble N-ethylmaleimide-sensitive factor attachment protein receptor; Syt1, synaptotagmin-1;

7. Figure 1: What is the meaning of letters “a,” “b,” “c,” “d”? The authors write: “Bars with different lowercase letters indicate significant difference (p < 0.05).” But which groups were compared to get the significant differences? The authors should clearly write this in the figure legends or indicate it in the figures. The same applies to all the figures.

Dear Reviewers and Editors,

We gratefully appreciate for your valuable suggestion. Data were analyzed by one-way ANOVA (Duncan's test) using SPSS (version 24.0). It was considered significant at P < 0.05. The a, b, c markers that appear in the figures are considered "significance markers." In our experiment, when the CG group is labeled as 'a' and the RSP group as 'b' or 'c', the distinct lowercase letter designations indicate a significant difference between the CG and RSP groups (P < 0.05). In contrast, when the CG group was labeled a and the RSP group was labeled a or ab there was no significant difference between the CG and RSP groups at this point (P > 0.05). Similarly, in the case of comparisons such as CG or RSP groups with other groups, simply observe the lowercase letters that are marked. Similar, only lowercase letters need to be observed to determine the difference between groups. We hope the revised manuscript meets your expectations. Your constructive comments are much appreciated.

8. Line 258: Another section “3.2”? This should be section “3.3”, as there is already a “3.2.”.

Dear Reviewers and Editors,

Thank you for the kind reminder, we have made the change. Thank you for your worthy suggestion.

9. Line 259: Grammar/typo: “Hippocampal neuronal damage and reduction as a marker of clinical depression.” Maybe the authors mean “Hippocampal neuronal damage and reduction is a marker of clinical depression.”?

Dear Reviewers and Editors,

Thank you for your nice suggestion and advice. With reference to your comments, the modifications have made our expression more precise. We sincerely hope that you are satisfied with the new changes.

10. Line 266: Typo: “RSP+SSA group This indicates that” should be “RSP+SSA group. This indicates that.”

Dear Reviewers and Editors,

Thank you for your careful review and constructive suggestions regarding our manuscript. We revised the manuscript by adding a period after RSP + SSA group. We apologize for our carelessness. Thank you once again for your valuable advice.

11. For Figure 3, the authors should quantify the number of positive cells in the hippocampus before claiming that “The findings indicate a significant increase in the number of positive cells within the hippocampus of mice in the RSP group when compared to the CG group” (Line 263-265). The same applies to Figure 3B. The author should quantify the HE staining results to show that there are “reduced neuronal cells in the hippocampus with structural abnormalities.” The same applies to Figure 4.

Dear Reviewers and Editors,

Thank you for your comments and providing good suggestions for our manuscript. We have visualized the proportion of FJB-positive cells in the hippocampus. In addition, in the HE section we have indicated the location of the lesion. The hippocampal cells at the lesion were abnormally arranged and reduced in number. In contrast, the hippocampal cells of the other groups of mice possessed a normal structure and the cells were tightly arranged. Visualization of ROS fluorescence intensity in the hippocampus is shown in Figure 4B. We hope will meet with your approval.

12. Figure 4A: The staining in Figure 4A is too faint to see. The author should quantify the signal to show the differences.

Dear Reviewers and Editors,

---

## [Decision Letter · Decision Letter 1]

16 Sep 2024

Saikosaponin A alleviates depressive-like behavior induced by reserpine in mice by regulating gut microflora and inflammatory responses

PONE-D-24-20060R1

Dear Dr. Yin,

We’re pleased to inform you that your manuscript has been judged scientifically suitable for publication and will be formally accepted for publication once it meets all outstanding technical requirements.

Kind regards,

Peng Zhong, Ph.D.

Academic Editor

PLOS ONE

Reviewers' comments:

Reviewer's Responses to Questions

**Comments to the Author**

1. If the authors have adequately addressed your comments raised in a previous round of review and you feel that this manuscript is now acceptable for publication, you may indicate that here to bypass the “Comments to the Author” section, enter your conflict of interest statement in the “Confidential to Editor” section, and submit your "Accept" recommendation.

Reviewer #1: All comments have been addressed

Reviewer #2: All comments have been addressed

2. Is the manuscript technically sound, and do the data support the conclusions?

Reviewer #1: Yes

Reviewer #2: Yes

3. Has the statistical analysis been performed appropriately and rigorously? 

Reviewer #1: Yes

Reviewer #2: Yes

4. Have the authors made all data underlying the findings in their manuscript fully available?

Reviewer #1: Yes

Reviewer #2: Yes

5. Is the manuscript presented in an intelligible fashion and written in standard English?

Reviewer #1: Yes

Reviewer #2: Yes

6. Review Comments to the Author

Reviewer #1: (No Response)

Reviewer #2: The revised version of manuscript is better than previous version. For one of my questions, the author should clarify the behavior tests were done in light or dark phase as rodent is nocturnal animals.

7. PLOS authors have the option to publish the peer review history of their article (what does this mean? ). If published, this will include your full peer review and any attached files.

**Do you want your identity to be public for this peer review?** For information about this choice, including consent withdrawal, please see our Privacy Policy .

Reviewer #1: No

Reviewer #2: No

---

## [Editor Report · Acceptance letter]

PONE-D-24-20060R1

PLOS ONE

Dear Dr. Yin,

I'm pleased to inform you that your manuscript has been deemed suitable for publication in PLOS ONE. Congratulations! Your manuscript is now being handed over to our production team.

Kind regards,

on behalf of

Dr. Peng Zhong

Academic Editor

PLOS ONE